# Cost-effectiveness of a radio intervention to stimulate early childhood development: protocol for an economic evaluation of the SUNRISE trial in Burkina Faso

Tom Palmer ![ORCID],[1] Abbie Clare,[2] Pasco Fearon,[1] Roy Head,[2] Zelee Hill ![ORCID] ,[1] Bassirou Kagone,[3] Betty Kirkwood,[4] Alexander Manu,[5] Jolene Skordis,[6] on behalf of the SUNRISE team

¹University College London, London, UK
²Development Media International, London, UK
³Development Media International, Ouagadougou, Burkina Faso
⁴LSHTM, London, UK
⁵London School of Hygiene and Tropical Medicine, London, UK, London, UK
⁶University College London Medical School, London, UK

**Correspondence to**
Tom Palmer; t.palmer@ucl.ac.uk

## ABSTRACT

**Introduction** Approximately 250 million children under 5 years of age are at risk of poor development in low-income and middle-income countries. However, existing early childhood development (ECD) interventions can be expensive, labour intensive and challenging to deliver at scale. Mass media may offer an alternative approach to ECD intervention. This protocol describes the planned economic evaluation of a cluster-randomised controlled trial of a radio campaign promoting responsive caregiving and opportunities for early learning during the first 3 years of life in rural Burkina Faso (*SUNRISE* trial).

**Methods and analysis** The economic evaluation of the *SUNRISE* trial will be conducted as a within-trial analysis from the provider's perspective. Incremental costs and health outcomes of the radio campaign will be compared with standard broadcasting (ie, 'do nothing' comparator). All costs associated with creating and broadcasting the radio campaign during intervention start-up and implementation will be captured. The cost per child under 3 years old reached by the intervention will be calculated. Incremental cost-effectiveness ratios will be calculated for the trial's primary outcome (ie, incremental cost per SD of cognitive gain). A cost-consequence analysis will also be presented, whereby all relevant costs and outcomes are tabulated. Finally, an analysis will be conducted to assess the equity impact of the intervention.

**Ethics and dissemination** The *SUNRISE* trial has ethical approval from the ethics committees of the Ministry of Health, Burkina Faso, University College London and the London School of Hygiene and Tropical Medicine. The results of the economic evaluation will be disseminated in a peer-reviewed journal and presented at a relevant international conference.

**Trial registration number** The *SUNRISE* trial was registered with ClinicalTrials.gov on 19 April 2019 (identifier: NCT05335395).

## STRENGTHS AND LIMITATIONS OF THIS STUDY

⇒ To our knowledge, this will be the first economic evaluation of a mass media campaign to improve early childhood development.
⇒ Estimates of the cost-effectiveness of the radio campaign as an intervention to improve early childhood development will help guide decision-makers in this setting.
⇒ An equity impact analysis will be conducted alongside the study.
⇒ Uncertainties may remain regarding long-term effectiveness and cost-effectiveness, particularly given uncertainty over whether intervention impact is sustained into adulthood.
⇒ Societal perspective analysis is not possible, as household costs associated with the intervention are not captured within planned data collection.

in low-income and middle-income countries.[1] An increased commitment to early childhood development (ECD) from international institutions resulted in the Nurturing Care Framework (NCF), launched during the 71st World Health Assembly in 2018.[2] The NCF focuses on the critical period from pregnancy to age 3, when brain development provides foundations for lifelong learning and physical and mental health.[3–5] Education outcomes in later life are dependent on ECD, meaning that developmental inequalities between young children are likely to persist and widen over the life course.[6 7]

The SUNRISE trial aims to address responsive caregiving and opportunities for early learning, two key components of the NCF,[2] by promoting physical, social, emotional and cognitive development. Accumulating evidence suggests interventions that aim to change caregiver behaviour, beliefs or

## INTRODUCTION

Approximately 250 million children under 5 years of age are at risk of poor development

practices can lead to improved ECD and that interventions promoting responsive caregiving may be particularly effective.[8–15] In Jamaica, the only low-income and middle-income context with available long-term evidence on future earnings impact, a psychosocial stimulation intervention delivered at age 9–24 months was associated with a 25% increase in earnings 20 years later.[16]

The success of the 'Reach Up and Learn' intervention in Jamaica has led to its adaptation and implementation across many diverse contexts globally. However, this approach commonly uses community health workers to provide information and materials to families in a face-to-face setting, which can be expensive, labour-intensive and challenging to deliver at scale.[8 17 18] Existing interventions promoting responsive caregiving have also tended to focus on the mother–child dyad,[19] while there have been calls to engage the wider family.[8 20] Mass media may offer an alternative approach to ECD intervention, which can deliver standardised messages to millions of people, multiple times per day. This may reach a wider variety of caregivers and stakeholders. Mass media is also potentially cost-effective at scale, as suggested by the results of a randomised controlled trial of a radio campaign targeting child survival in rural Burkina Faso.[21]

Although cost and affordability are crucial elements of successful scale-up of ECD interventions,[22] there remains little primary evidence on the cost and cost-effectiveness of ECD interventions.[23 24] Existing economic evaluations of ECD interventions include studies in China,[25] Pakistan[26] and Vietnam.[27] Only one known study provides relevant cost-effectiveness analysis from sub-Saharan Africa. An evaluation of a parenting group intervention in Kenya found that provider costs per child were US$119, resulting in an incremental cost-effectiveness ratio (ICER) of a 0.37 SD improvement in cognition per US$100 invested.[28] The authors also modelled potential future benefits of early cognitive gains through increased lifetime earnings, estimating a benefit-cost ratio of 15.5 (ie, a US$15.5 increase in future lifetime earnings for every US$1 spent on the intervention). This suggests that the benefits of similar ECD interventions (in similar contexts) are likely to outweigh their costs. However, the calculated provider costs per child in that study were substantially above annual per capita health expenditure in Kenya.[29] This may make such interventions unaffordable in comparable resource-constrained settings, precluding the widespread rollout of ECD interventions unless lower cost and scalable interventions are found.

Mass media approaches are potentially scalable, with a low cost per person reached. To our knowledge, the present study is the first economic evaluation of a mass media campaign to improve ECD, and therefore will provide crucial evidence informing affordability and feasibility at the scale of similar interventions. It is also one of very few studies providing cost and cost-effectiveness evidence for any form of ECD intervention tested in sub-Saharan Africa, and the first of its kind in Burkina Faso.

## The SUNRISE trial

The SUNRISE trial is a two-arm cluster-randomised controlled trial in rural Burkina Faso. The trial evaluates the impact of a radio campaign on ECD outcomes. The radio campaign promotes responsive caregiving and opportunities for early learning during the first 3 years of life. The SUNRISE trial is described in detail elsewhere.[30] This article outlines only the methodology used in the economic evaluation of the trial.

## OBJECTIVES

The objectives of the economic evaluation of the SUNRISE radio campaign are to:
1. Estimate the total cost of delivering the intervention from the provider's perspective.
2. Estimate the incremental cost and cost-effectiveness of the intervention compared with 'doing nothing' from the provider's perspective.
3. Measure the equity impact of the intervention.

## METHODS
### Study design, setting and population

Radio is the dominant form of media in Burkina Faso. According to the most recent, nationally representative data, 60% of households possess a radio (compared with 35% TV ownership) and 44% listen to the radio at least weekly.[31] Only 20% of the population have electricity access,[31] meaning that battery or solar-powered radios or mobiles are widely used.

The unit of randomisation in the SUNRISE trial was local FM radio stations and their catchment areas, with eight radio stations randomised to broadcast the SUNRISE campaign and seven to act as controls. Each of these clusters was required to meet the following criteria: there are at most two dominant radio stations and no discernible broadcast from stations in neighbouring clusters; they are located outside of the large cities of Ouagadougou or Bobo-Dioulassou, and they are not subject to significant security risks. These criteria were used to minimise the risk of contamination between clusters, such that FM radio stations with geographically distinct catchment areas were selected for the trial.

The target population of the radio campaign includes all residents living within the radio station catchment areas. Trial participants were recruited from evaluation areas within each cluster. Criteria for evaluation areas include proximity to the radio station, high signal strength based on a motorbike survey and high listenership based on a media survey. Towns, villages within 5 km of towns, villages on the national electricity grid and villages with populations exceeding 5000 (as they are likely to be a priority for the national electrification programme) are excluded from the evaluation areas. These criteria were used to ensure local radio is the main communication channel, with minimal audiences lost to television. All children born between January and June 2022 in each

evaluation area, and their families, were enrolled in the trial, and will be followed longitudinally until they reach 30 months of age. Exclusion criteria were babies with major congenital disorders, babies not living with their mothers and babies with mothers who were not capable of participating in assessments. Formal introductions and permissions were sought from community leaders in all evaluation areas using a standardised protocol and all participants provided informed consent. The trial cluster randomised design was explained and they were made aware that the SUNRISE radio campaign promoting responsive parent–child interactions, play, child-directed language and praise will be broadcast daily through local radio stations in eight of the trial clusters and that radio stations in the other seven areas will continue to broadcast their usual range of programmes. It was also explained that the areas that get the SUNRISE broadcasts will be decided at random like a lottery and that the data collected from participants will help to decide whether the SUNRISE campaign makes a difference to child development outcomes.

Radio listenership will be evaluated quantitatively every 3 months throughout the duration of the trial, including whether a radio or mobile phone was used. Data on device ownership will be collected at trial enrolment. Exposure to other sources of information will also be asked about in the qualitative process evaluation.

Data will also be collected from community leaders and resident fieldworkers concerning any child health or early child development interventions being implemented in the trial evaluation areas.

## Intervention

The SUNRISE radio campaign is based on two principles: saturation and interactivity. The campaign uses 60 s advertisements that are broadcast 10 times per day, 365 days per year for 3 years, on local radio stations in the intervention clusters. Advertisement scripts were based on extensive formative research to develop messages and approaches that are feasible, well-targeted and appropriate for the environmental and cultural context. Additionally, 60 min programmes are broadcast twice-weekly on weekday evenings, incorporating dramas, real-life testimonials and practical 'how to' advice, with space for listeners to phone in to discuss the advertisements and ask questions. In the control clusters, the local radio stations continue with their usual broadcasting schedule.

Formative research was conducted to investigate barriers and facilitators related to responsive caregiving practices that are linked to improved ECD outcomes. Findings from this research were then combined with an ECD curriculum developed by subject experts and the resulting 'message briefs' formed the basis of content development. The content promotes cognitive development, responsive caregiving and nurturing care of children aged 0–3 years. All content is tested with target audiences during the production cycle, and includes scripted dramas, real-life testimony and modelling of responsive parent–child interactions.

The radio campaign will run for 3 years from October 2021 until September 2024; the target population in the intervention evaluation areas will be approximately 8000 children aged less than 3 years, with approximately 800 babies born between January and June 2022 enrolled in the trial for outcome data collection.

## Measurement of child development outcomes

The trial surveillance system comprises regular 3-monthly home visits by fieldworkers to all enrolled households to collect data. The primary impact outcome is the age-adjusted development for age z-score on the Global Scales for Early Development Long Form, which will be assessed at 30–32 months of age. The secondary impact outcomes of the trial include the Caregiver Reported Early Development Instruments Long Form scores measured 6-monthly from 6 to 8 months of age until 30–32 months of age, and the Communicative Development Inventories score at 21–23 months and 27–29 months, which aims to assess early language development. Intermediate home environment outcomes will also be measured. The z-standardised sum score for observed responsiveness from the National Institute of Child Health and Human Development sensitivity scales, assessed at 15–17 months of age, is the primary intermediate outcome and the Home Observation for Measurement of the Environment—(Infant/Toddler) score assessed at 24–26 months is the secondary intermediate outcome.

Trial findings will be reported according to the Consolidated Standards of Reporting Trials guidelines for cluster randomised trials. Analyses will be intention-to-treat and include all data from trial children and their families, regardless of their exposure to the radio campaign. Random-effects linear regression models using individual-level data will be used to adjust for the clustered design and any imbalances between intervention and control arms. Effect sizes will be presented as standardised mean differences with 95% CIs. The data analysis plan will be agreed upon with the trial technical steering and data management and ethics committees prior to the end of recruitment.

## Identification, measurement and valuation of resource use

Cost and cost-effectiveness analyses will be conducted from the provider perspective, accounting for the costs incurred by all providers in the provision of the intervention. An overview of cost data is presented in table 1.

Provider costs include the costs incurred during intervention start-up and implementation and will capture all costs associated with creating and broadcasting the advertisements. Provider costs will be incurred by Development Media International (DMI), Innovation for Poverty Action (IPA), the London School of Hygiene and Tropical Medicine (LSHTM) and University College London (UCL). DMI will incur costs for broadcasting the campaign. Costs for developing campaign materials, including carrying

**Table 1** Description of costs and data sources

| Cost category | | Hypothesis/potential cost impact | Proposed data source(s) |
|---|---|---|---|
| **Provider** | Intervention costs for designing, starting up and implementing the radio campaign (**Programme costs**) | ► Direct increase in costs in the short-term due to equipment, staff costs, training costs, travel expenses, etc. <br>► Indirect costs include the opportunity cost of donated items, volunteer staff time. <br>► Capital expenditure related to content creation. <br>► Cost of airtime for radio content. | ► Project financial records. <br>► Project staff time sheets. <br>► Project staff interviews. <br>► Project records on volunteer involvement. |

out formative research, will be incurred by DMI London, DMI Burkina, IPA and UCL. LSHTM will incur costs for advising on campaign content and approach. Data for these costs will be based on financial project accounts, prospectively collected from each institution.

Provider costs will be entered into a customised Microsoft Excel tool on an annual basis. The tool is divided into different sections based on cost components, such as staff, materials, capital and joint costs. Joint costs include administration, overheads and other costs that are shared across project components. These costs will be allocated to programme components based on both staff time-use and key informant interviews with representatives of each institution. Research costs-related strictly to trial evaluation will be excluded from the economic analysis, although contributions from the research team towards campaign design will be captured within start-up costs. Where relevant, financial costs will be converted to economic costs. For example, any donated goods predating the project will be assigned a current market value in the costing tool.

### Economic evaluation

The economic evaluation of the SUNRISE intervention will fully cost, from the provider's perspective, a large-scale intervention to promote early child development. The analysis will be conducted as a within-trial analysis, using the trial results. Estimating provider costs will enable both cost-effectiveness analysis and cost-consequence analysis. The cost per child under 3 years old reached by the intervention will be calculated.

ICERs (ie, mean difference in cost between the intervention and control arms, divided by mean difference in effect) will be calculated for the primary outcome (ie, cost per SD of cognitive gain), provided an impact of the intervention on this outcome is detected. A cost-consequence analysis will also be presented, whereby all relevant costs and outcomes are tabulated, allowing policymakers to easily compare incremental costs and incremental effects of the SUNRISE radio campaign.

Costs will be adjusted for inflation to 2024 base year values using the Consumer Price Index for Burkina Faso and presented in current prices in both West African CFA franc (CFA) and international dollars (INT$). Conversion to INT$ will use the purchasing power parity conversion factor for Burkina Faso. Costs will be discounted at the standard annual discount rate of 3% in the base-case analysis. All costs will be assessed over the full-time horizon of the trial (March 2021 to May 2025), including the development of the campaign and preparation phase of the trial. One-way sensitivity analyses will be used to explore the impact of changes in intervention effectiveness (95% CI), intervention cost (±25%) and discount rate (0%–6%) on the cost-effectiveness estimates.

Finally, should the intervention have a positive impact on the primary or secondary outcomes, a fiscal space analysis will be conducted to assess the feasibility of the government allocating resources to a national rollout of the campaign.[32] Total provider costs will form the basis for affordability estimates, calculated as a percentage of national gross domestic product. The total cost of a fully scaled programme will also be compared with current health and education spending in context. Given that intervention effectiveness estimates are derived from rural areas as described above, it is possible that outcomes will not be generalisable to urban areas. To account for this, a range of scenario analyses will be conducted to explore the impact of different assumptions regarding intervention effectiveness on estimates of cost-effectiveness. Scenario analyses will also account for likely differences in costs, given that costs are likely to be substantially lower at scale, due to economies of scale in cost components such as intervention content development.

### Equity impact

The equity impact or benefit incidence of the intervention will also be measured, given that poverty is associated with lower levels of cognitive development.[3 33] This analysis will evaluate how the benefits of the intervention are distributed across different socioeconomic groups. All primary and secondary trial outcomes will be analysed according to the socioeconomic status of the household at recruitment into the trial, as measured using a Multidimensional Poverty Index (MDPI). The MDPI will be derived using the Alkire and Foster method[34] and will incorporate indicators of maternal and paternal levels of education, and household living standards. The MDPI accounts for both monetary and non-monetary dimensions of deprivation, enabling differentiation between population groups who may seem homogeneously asset or cash poor. MDPI quintiles will be used to analyse benefit incidence

across socioeconomic groups. The random effects model described above for analysis of trial outcomes will be used, with the addition of a factorial interaction of MDPI quintile and intervention effect.

## Data management

Data protection and confidentiality procedures will be in keeping with Good Clinical Practice and the General Data Protection Regulation 2018. Research Electronic Data Capture(REDCap) will be used for data collection which is a secure web application for building and managing online surveys and databases, and which will have in-built range and consistency checks. All data will be collected on secure, password-protected handheld tablets within an encrypted database, with the data transferred daily to a secure password-protected cloud server. Encrypted transfer of pseudonymised quantitative data to LSHTM will take place monthly.

The following systematic process of cultural adaptation will be carried out for all instruments: Translation of the assessment instruments into the main local languages spoken by residents in the trial clusters, and adaptation of the testing materials for the local context; ensuring technical equivalences; cognitive interviews with respondents and project staff (field research); modifications of translated versions, based on the field research; pretesting, including further modification; training of assessors, establishing inter-rater reliability; and pilot-testing, including testing of standard operating procedures.

Outcome and socio-demographic data from trial participants will be curated into a single anonymised Stata .dta file, including newly generated participant ID and cluster codes, variable labels and defined missing values. All potentially identifiable data, including name and location, will be removed from the data set. Detailed data dictionaries will be produced to accompany the data set. All data processing will be documented in annotated Stata analysis.do scripts. No additional individual-level data will be collected for the economic analysis.

## Patient and public involvement

Patients and public are not involved in the design or conduct of this economic evaluation study. Any individual-level data used in the economic evaluation will be collected only after obtaining voluntary, informed consent from participants.

## Ethics and dissemination

The *SUNRISE* trial has ethical approval from the ethics committees of the Ministry of Health, Burkina Faso, UCL and the LSHTM. The results of the economic evaluation will be disseminated in a peer-reviewed journal and presented at a relevant international conference.

**Contributors** TP and JS-W conceived and designed this economic evaluation study. BKi, PF, ZH and AM developed the SUNRISE trial evaluation design. AM, PF and BKi are responsible for trial data collection and statistical analysis of impact evaluation, and ZH for the process evaluation. AC, RH and BKa designed the radio campaign with input from ZH and PF. TP drafted the economic evaluation protocol and all authors contributed to revised drafts. All authors read and approved the final manuscript.

**Funding** The SUNRISE trial is funded by the NIHR–Wellcome Partnership for Global Health Research Collaborative Award, with separate contracts to LSHTM & DMI (LSHTM: 215492/Z/19/Z; DMI: 215492/A/19/Z). Additional funding to support radio campaign costs was received by DMI from the Light Foundation.

**Competing interests** None declared.

**Patient and public involvement** Patients and/or the public were not involved in the design, or conduct, or reporting, or dissemination plans of this research.

**Patient consent for publication** Not applicable.

**Provenance and peer review** Not commissioned; externally peer reviewed.

**Data availability statement** N/A.

**ORCID iDs**
Tom Palmer http://orcid.org/0000-0002-9526-0045
Zelee Hill http://orcid.org/0000-0002-2614-8877

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
