## [Reviewer comments · BMJ Open]

ARTICLE DETAILS

TITLE (PROVISIONAL)	Cost Effectiveness of a Radio Intervention to Stimulate Early Childhood Development: Protocol for an Economic Evaluation of the SUNRISE trial in Burkina Faso
AUTHORS	Palmer, Tom; Clare, Abbie; Fearon, Pasco; Head, Roy; Hill, Z; Kagone, Bassirou; Kirkwood, Betty; Manu, A; Skordis-Worrall, Jolene

VERSION 1 – REVIEW

REVIEWER	Moretti, Myla The Hospital for Sick Children, University of Toronto, Clinical Trials Unit
REVIEW RETURNED	26-Dec-2023

GENERAL COMMENTS	General comments to the authors: Thank you for the opportunity to review this interesting and novel proposed work. Table of acronyms is needed and all acronyms should be spelled out in full the first time they appear in the document. Specific Comments: Abstract, Methods and analysis: in the parenthesis “i.e. cost per “ should be...”incremental cost per.....” UCL and LSHTM appear for the first time in the abstract, please spell out. Introduction: The background, existing evidence and rationale is well laid out and described appropriately. Objectives: using only provider perspective will not account for family/caregiver costs which could results in not capturing, or shifting, of costs from the provider to other sectors or individuals. This should be acknowledged upfront as a significant limitation of the proposed work. Methods, Study design, setting and population: It is not clear if households had access to the radio transmissions. Would radio typically be the only external media available and would it be available in all households normally in these regions? How do the investigators plan to measure or account for other media or local interventions that may be impacting the outcomes measured? A sentence or two to elaborate on that is warranted. Methods, Intervention: A brief description of the content of the SUNRISE intervention and its development is needed. What was the expertise of those responsible for developing the content?
---

	What is it advertising? What is it intended to do, or promote? Was the content itself educational or was it guiding parents to seek out resources or did it encourage specific action? Did members of the communities know there was an intervention happening? The intervention should be described more fully here. Methods: Some explanation of the data management is needed and should include a description of data cleaning, handling missing values, detecting, and handling outliers, data validation etc. Methods, Economic Evaluation: More detail should be included with respect to an analytical plan, notably the sensitivity analysis plan. For example, how will the ranges be established around uncertain variables? Equity Impact: The statistical plan should, a priori, describe the type of model being used and the covariates to be included in the model. What is the source of the data used in the equity impact? How do the investigators derive the MDPI (education living standards etc) or any other socioeconomic variables?
--	---

REVIEWER	Labadie, Guilhem UNICEF, Immunization
REVIEW RETURNED	28-Dec-2023

GENERAL COMMENTS	 - Clarify the time frame and time horizon, as it is not clear how it aligns with the cohort and intervention you define. Also how do you address increase in unit costs if children not included in the cohort (a few months older) would still benefit from the intervention? - Explain why you exclude babies not living with their mother, especially as you make an equity analysis. For instance HIV orphans living with other parents should also benefit from these ECD intervention, especially if your objective is to measure equity and children with less access to ECD. This doesn't make sense if your objective is to have an equity analysis, especially at household level ("Household-level data"). Do you know how many babies are you excluding? This may increase your unit costs too - Clarify risk of contamination of effects - The intervention seems to measure cost effectiveness for rural remote areas but you mention "assess the feasibility of the government allocating resources to a national roll out of the campaign". Clarify if it would be just a national roll out for remote rural areas, if not the cost effectiveness measured may not apply to other areas, specifically with TV access etc. as you describe them. - Clarify how you will measure MDPI over the 3 years: initially? at the end? how will you address changes over the years in MDPI? - Clarify how you will assign Capital vs recurrent or operating costs - Clarify if any other ECD intervention will be run in the areas under review (different policies, cultures, etc.?) - Will this intervention displace other health campaigns (e.g., promotion of vaccination, etc.) and will you measure the negative effects? - Give an estimate of the scale of the population targeted (is it 50 children or 5000) Very interesting!
---

VERSION 1 – AUTHOR RESPONSE

Reviewer: 1

Dr. Myla Moretti, The Hospital for Sick Children, University of Toronto

Comments to the Author:

General comments to the authors:

Thank you for the opportunity to review this interesting and novel proposed work.

Table of acronyms is needed and all acronyms should be spelled out in full the first time they appear in the document.

Thank you. As far as we understand, use of a table of acronyms is not compatible with the editorial style of BMJ Open. We have checked the first use of acronyms throughout the document and amended the manuscript to include a full spelling where necessary.

Specific Comments:

Abstract, Methods and analysis:

In the parenthesis “i.e. cost per “ should be...”incremental cost per.....”

This change has been made in the manuscript.

UCL and LSHTM appear for the first time in the abstract, please spell out.

This change has been made in the manuscript.

Introduction:

The background, existing evidence and rationale is well laid out and described appropriately.

Thank you for your feedback.

Objectives:

using only provider perspective will not account for family/caregiver costs which could results in not capturing, or shifting, of costs from the provider to other sectors or individuals. This should be acknowledged upfront as a significant limitation of the proposed work.

We have added the following limitation to the “Strengths and limitations” section:

“Societal perspective analysis is not possible, as household costs associated with the intervention are not captured within planned data collection”

Methods, Study design, setting and population:

It is not clear if households had access to the radio transmissions. Would radio typically be the only external media available and would it be available in all households normally in these regions? How do the investigators plan to measure or account for other media or local interventions that may be impacting the outcomes measured? A sentence or two to elaborate on that is warranted.

The “Study design, setting and population” section of the manuscript has been amended to include the following introduction on national radio coverage:

“Radio is the dominant form of media in Burkina Faso. According to the most recent, nationally representative data (the 2021 DHS survey), 60% of households possess a radio (compared with 35% TV ownership) and 44% listen to the radio at least weekly [ref]. Only 20% of the population have electricity access [ref], meaning that radio or solar-powered radios or mobile phones are widely used.” As explained in the protocol, evaluation villages were selected to fall outside of a 5km radius from the main cluster town, to reduce the likelihood of sampling households on the electricity grid (and therefore more likely to have higher television access). The following clarification regarding exposure monitoring has been added to the manuscript:

“Radio listenership will be evaluated quantitatively every 3 months throughout the duration of the trial, including whether a radio or mobile phone was used. Data on device ownership will be collected at

trial enrolment. Exposure to other sources of information will also be asked about in the qualitative process evaluation.”

Methods, Intervention: A brief description of the content of the SUNRISE intervention and its development is needed. What was the expertise of those responsible for developing the content? What is it advertising? What is it intended to do, or promote? Was the content itself educational or was it guiding parents to seek out resources or did it encourage specific action? Did members of the communities know there was an intervention happening? The intervention should be described more fully here.

The following additional detail on the intervention has been added to the “Intervention” section of the manuscript:

“Formative research was conducted to investigate barriers and facilitators related to responsive caregiving practices that are linked to improved ECD outcomes. Findings from this research were then combined with an ECD curriculum developed by subject experts and the resulting “message briefs” formed the basis of content development. The content promotes cognitive development, responsive caregiving and nurturing care of children aged 0-3 years. All content is tested with target audiences during the production cycle, and includes scripted dramas, real-life testimony and modelling of responsive parent-child interactions.”

The following clarification was added regarding communities knowing whether an intervention was happening in the Study Design Section.

“Formal introductions and permissions were sought from community leaders in all evaluation areas using a standardised protocol and all participants provided informed consent. The trial cluster randomised design was explained and they were made aware that the SUNRISE radio campaign promoting responsive parent-child interactions, play, child-directed language and praise will be broadcast daily through local radio stations in 8 of the trial clusters and that radio stations in the other 7 areas will continue to broadcast their usual range of programmes. It was also explained that the areas that get the SUNRISE broadcasts will be decided at random like a lottery and that the data collected from participants will help to decide whether the SUNRISE campaign makes a difference to child development outcomes.”

Methods:

Some explanation of the data management is needed and should include a description of data cleaning, handling missing values, detecting, and handling outliers, data validation etc.

The following section on data management has been added to the manuscript:

“Data protection and confidentiality procedures will be in keeping with Good Clinical Practice (GCP) and the General Data Protection Regulation 2018. REDCap will be used for data collection which is a secure web application for building and managing online surveys and databases, and which will have in-built range and consistency checks. All data will be collected on secure, password protected handheld tablets within an encrypted database, with the data transferred daily to a secure password protected cloud server. Encrypted transfer of pseudonymized quantitative data to LSHTM will take place monthly.

The following systematic process of cultural adaptation will be carried out for all instruments:

Translation of the assessment instruments into the main local languages spoken by residents in the trial clusters, and adaptation of the testing materials for the local context; Ensuring technical equivalences; Cognitive interviews with respondents and project staff (field research); Modifications of translated versions, based on the field research; Pretesting, including further modification; Training of assessors, establishing inter-rater reliability; and Pilot-testing, including testing of standard operating procedures.

Outcome and sociodemographic data from trial participants will be curated into a single anonymised STATA .dta file, including newly generated participant ID and cluster codes, variable labels, and defined missing values. All potentially identifiable data, including name and location, will be removed from the dataset. Detailed data dictionaries will be produced to accompany the dataset. All data

processing will be documented in annotate STATA analysis .do scripts. No additional individual-level data will be collected for the economic analysis.”

Methods, Economic Evaluation:

More detail should be included with respect to an analytical plan, notably the sensitivity analysis plan. For example, how will the ranges be established around uncertain variables?

The following paragraph has been added to the section on Measurement of child development outcomes:

“Trial findings will be reported according to the CONSORT guidelines for cluster randomized trials. Analyses will be intention to treat and include all data from trial children and their families, regardless of their exposure to the radio campaign. Random-effects linear regression models using individual-level data will be used to adjust for the clustered design and any imbalances between intervention and control arms. Effect sizes will be presented as standardized mean differences with 95% confidence intervals (CI). The data analysis plan will be agreed with the trial technical steering and data management and ethics committees prior to the end of recruitment.”

The text regarding sensitivity analysis in the Economic evaluation section has been amended as follows:

“One-way sensitivity analyses will be used to explore the impact of changes in intervention effectiveness (95% confidence interval), intervention cost (+/-25%) and discount rate (0%-6%) on the cost-effectiveness estimates.”

Equity Impact:

The statistical plan should, a priori, describe the type of model being used and the covariates to be included in the model. What is the source of the data used in the equity impact? How do the investigators derive the MDPI (education living standards etc) or any other socioeconomic variables? The text regarding the MDPI has been amended in the manuscript, and now reads as follows:

“All primary and secondary trial outcomes will be analysed according to the socioeconomic status of the household at recruitment into the trial, as measured using a multidimensional poverty index (MDPI). The MDPI will be derived using the Alkire and Foster method (Alkire and Foster, 2011) and will incorporate indicators of maternal and paternal levels of education, and household living standards. The MDPI accounts for both monetary and non-monetary dimensions of deprivation, enabling differentiation between population groups who may seem homogeneously asset or cash poor. MDPI quintiles will be used to analyse benefit incidence across socioeconomic groups. The random effects model described above for analysis of trial outcomes will be used, with the addition of a factorial interaction of MDPI quintile and intervention effect.”

Reviewer: 2

Dr. Guilhem Labadie, UNICEF

Comments to the Author:

- Clarify the time frame and time horizon, as it is not clear how it aligns with the cohort and intervention you define.

The following clarification has been added to the Methods section:

“All costs will be assessed over the full time horizon of the trial (March 2021-May 2025), including the development of the campaign and preparation phase of the trial.”

Although benefits may be observed later in life, potentially supporting a longer time horizon, longer term follow-up is needed to determine whether intervention benefits are sustained. Modelling approaches could be the subject of future work, but this also introduces substantial uncertainty. Also how do you address increase in unit costs if children not included in the cohort (a few months older) would still benefit from the intervention?

It is conceivable that older children might benefit from the intervention, but this is not captured in our evaluation data. In which case, we may underestimate intervention cost-effectiveness. Household costs will not be captured in this study, and the analysis will be from the provider perspective.

- Explain why you exclude babies not living with their mother, especially as you make an equity analysis. For instance HIV orphans living with other parents should also benefit from these ECD intervention, especially if your objective is to measure equity and children with less access to ECD. This doesn't make sense if your objective is to have an equity analysis, especially at household level ("Household-level data"). Do you know how many babies are you excluding? This may increase your unit costs too

Please note, HIV prevalence remains low so that in this setting it is rare to find HIV orphans. We have excluded children whose mothers have died since they may be cared for by a variety of carers such as grandmothers, aunts, in-laws etc. Consistency of measurement may therefore be affected if a more diverse groups of carers is included as trial outcomes may be collected from different carers at different times during the trial. It is worth noting also that these babies are only excluded from the evaluation sample, and may still benefit from the intervention.

- Clarify risk of contamination of effects

The following clarification has been added to the "Study design, setting and population" section of the manuscript:

"These criteria were used to minimise the risk of contamination between clusters, such that FM radio stations with geographically distinct catchment areas were selected for the trial."

While we acknowledge contamination is possible through migration, this will be captured through planned trial data collection.

- The intervention seems to measure cost effectiveness for rural remote areas but you mention "assess the feasibility of the government allocating resources to a national roll out of the campaign". Clarify if it would be just a national roll out for remote rural areas, if not the cost effectiveness measured may not apply to other areas, specifically with TV access etc. as you describe them. Thank you for highlighting this important point. The trial takes place in rural areas to ensure non-overlapping signals of radio stations, providing the opportunity for proof of concept. As mentioned above, the following clarification has been added to the "Study design, setting and population" section of the manuscript:

"These criteria were used to minimise the risk of contamination between clusters, such that FM radio stations with geographically distinct catchment areas were selected for the trial. In practice, only rural areas are able to meet these criteria."

If the intervention is found to be effective, it will be easier to implement in urban areas with better radio coverage. However, we agree that generalisability of effectiveness estimates may be a concern. The following clarification has been added to the economic evaluation section:

"Given that intervention effectiveness estimates are derived from rural areas as described above, it is possible that outcomes will not be generalisable to urban areas. To account for this, a range of scenario analyses will be conducted to explore the impact of different assumptions regarding intervention effectiveness on estimates of cost-effectiveness. Scenario analyses will also account for likely differences in costs, given that costs are likely to be substantially lower at scale, due to economies of scale in cost components such as intervention content development."

- Clarify how you will measure MDPI over the 3 years: initially? at the end? how will you address changes over the years in MDPI?

The indicators used to construct the MDPI will be measured at baseline. We have amended the manuscript to reflect this. It may also be possible to collect this data at endline, but this will depend on available resources.

- Clarify how you will assign Capital vs recurrent or operating costs

Provider costs are categorised into cost components, including staff, materials, capital and joint costs. Costs will also be summarised by implementation phase (i.e. start-up or implementation phase). Typically, capital expenditure would be expected to be incurred during the start-up phase.

- Clarify if any other ECD intervention will be run in the areas under review (different policies, cultures, etc.?)

The following has been added to the study design section:

“Data will also be collected from community leaders and resident fieldworkers concerning any child health or early child development interventions being implemented in the trial evaluation areas.”

- Will this intervention displace other health campaigns (e.g., promotion of vaccination, etc.) and will you measure the negative effects?

The intervention will not displace other health campaigns.

- Give an estimate of the scale of the population targeted (is it 50 children or 5000)

The following has been added to the section on Intervention:

“The radio campaign will run for 3 years from October 2021 until September 2024; the target population in the intervention evaluation areas will be approximately 8,000 children aged less than 3 years, with approximately 800 babies born between January and June 2022 enrolled into the trial for outcome data collection.”